# Evaluation of the Antimicrobial and Antivirulent Potential of Essential Oils Isolated from *Juniperus oxycedrus* L. ssp. *macrocarpa* Aerial Parts

**DOI:** 10.3390/microorganisms10040758

**Published:** 2022-03-31

**Authors:** Gabriella Spengler, Márió Gajdács, Matthew Gavino Donadu, Marianna Usai, Mauro Marchetti, Marco Ferrari, Vittorio Mazzarello, Stefania Zanetti, Fruzsina Nagy, Renátó Kovács

**Affiliations:** 1Department of Medical Microbiology, Albert Szent-Györgyi Health Center and Albert Szent-Györgyi Medical School, University of Szeged, Semmelweis Utca 6, 6725 Szeged, Hungary; spengler.gabriella@med.u-szeged.hu; 2Department of Oral Biology and Experimental Dental Research, Faculty of Dentistry, University of Szeged, Tisza Lajos krt. 63, 6720 Szeged, Hungary; gajdacs.mario@stoma.szote.u-szeged.hu; 3Hospital Pharmacy, Azienda Ospedaliero Universitaria di Sassari, 07100 Sassari, Italy; 4Department of Biomedical Sciences, University of Sassari, 07100 Sassari, Italy; dr.marcoferrari@gmail.com (M.F.); vmazza@uniss.it (V.M.); zanettis@uniss.it (S.Z.); 5Department of Chemistry and Pharmacy, University of Sassari, 07100 Sassari, Italy; dsfusai@uniss.it; 6Institute of Biomolecular Chemistry (CNR), Li Punti, 07100 Sassari, Italy; mauro.marchetti@cnr.it; 7Department of Medical Microbiology, Faculty of Medicine and Pharmacy, University of Debrecen, Nagyerdei krt. 98, 4032 Debrecen, Hungary; nagyfruzsina0429@gmail.com (F.N.); kovacs.renato@med.unideb.hu (R.K.); 8Doctoral School of Pharmaceutical Sciences, University of Debrecen, 4032 Debrecen, Hungary

**Keywords:** antibacterial, antifungal, essential oil, *Juniperus oxycedrus*, Candida, *C. auris*, efflux pump, biofilm, multidrug resistance, MDR

## Abstract

As a consequence of the worsening situation with multidrug-resistant (MDR) pathogens and a disparity in the commercialization of novel antimicrobial agents, scientists have been prompted to seek out new compounds with antimicrobial activity from a wide range of sources, including medicinal plants. In the present study, the antibacterial, antifungal, anti-virulence, and resistance-modulating properties of the essential oil from the Sardinian endemic *Juniperus oxycedrus* L. ssp. *macrocarpa* aerial parts were evaluated. The GC/MS analysis showed that the main compounds in the oil were α-pinene (56.63 ± 0.24%), limonene (14.66 ± 0.11%), and β-pinene (13.42 ± 0.09%). The essential oil showed potent antibacterial activity against Gram-positive bacteria (0.25–2 *v/v*%) and *Salmonella* spp. (4 *v/v*%). The strongest fungicidal activity was recorded against *Candida auris* sessile cells (median FICI was 0.088) but not against *C. albicans* biofilms (median FICI was 1). The oil showed potent efflux pump inhibitory properties in the case of *Staphylococcus aureus* and *Escherichia coli*. The therapeutic potential of Juniperus may be promising for future more extensive research and in vivo tests to develop new drugs against antibiotic and antifungal resistance.

## 1. Introduction

The introduction and clinical use of various classes of antimicrobials have become one of the most important hallmarks of modern healthcare, leading to a significant benefit in survival rates and quality of life for patients affected by infectious ailments [1]. The emergence of drug resistance in these pathogens over the last several decades has increasingly become a serious issue worldwide, with multidrug-resistant (MDR) organisms becoming progressively more common, owing to the indiscriminate use of the commercially-available medications to treat infectious illnesses in human and animal medicine [2,3]. While previously, this issue was confined to resistant bacteria, at present, the term antimicrobial resistance (AMR) is aptly used, referring to the development of resistance to viruses, fungi, protozoa, and parasites, in addition to MDR bacteria [4]. By the 21st century, AMR has emerged as one of the leading causes of death, with patients in low- and middle-income countries being disproportionally affected [5,6].

Because of a worsening AMR situation and a disparity in the commercialization of novel antimicrobial agents, scientists have been led to seek out new compounds with antimicrobial activity from a wide range of sources, including medicinal plants rich in novel antimicrobial chemotherapeutic agents [7,8,9]. In fact, around 20–50% of recently authorized small-molecule drugs have been developed from compounds of plant origins [10]. With the increasing relevance of ethnopharmacology and ethnomedicine, essential oils (EOs) have been rediscovered as highly respected therapeutic aids for their high bactericidal and bacteriostatic power, i.e., for their unequivocal ability to kill pathogenic bacteria or to inhibit their multiplication without interfering with the normal microbiota of the host when administered [11,12]. The antimicrobial potency of EOs has been known for many years; in particular, the EOs of *Melaleuca alternifolia* L., *Thymus vulgaris* L., *Mentha piperita* L., and *Rosmarinus officinalis* L. and other natural drugs were and are used for the treatment of a wide range of bacterial, fungal, and viral infections [13,14,15].

*Juniperus* is the largest genus in the *Cupressaceae* family in terms of the number of species; it is characterized by fleshy cones with hard-shelled seeds, which is an adaptation to avian seed dispersal and has traditionally been divided into three distinct sections or subgenera [16]. *Juniperus oxycedrus* L. (*Cupressaceae*) (plum juniper, cada, cade juniper, prickly juniper, red-berry juniper) is a small tree or shrub native to the Mediterranean basin, ranging from Morocco and eastern Portugal to the western Caucasus, growing on a multitude of rocky sites ranging from 0 to 1600 m elevation above sea level [17]. *J. oxycedrus* L.—as mentioned by the Flora Europea—has three subspecies, including subsp. *badia* (H. Gay) Debeaux; subsp. *Oxycedrus*, subsp. *macrocarpa* (Sm.) Ball. The false fruits of *Juniperus*, the female cones—mistakenly referred to as “berries”—are mainly utilized in European cuisine as a spice, i.e., to give a strong, clear flavor to meat recipes in Northern European and especially Scandinavian cuisine [18]. In traditional Sardinian medicine, *J. oxycedrus* L. ssp. *macrocarpa* berries have widely been used to treat the common cold, gastrointestinal disorders, calcinosis in joints, hemorrhoids, and urinary inflammations and as an expectorant in coughs, a hypoglycemic, and a diuretic to pass kidney stones; in addition, the berries and leaves are applied externally for parasitic disease [19,20]. *J. oxycedrus* leaves, resin, bark, and berry extracts were found to prevent infections by a variety of microorganisms [21,22].

The aim of the present study was to evaluate the Sardinian endemism of *Juniperus oxycedrus* L. ssp. *macrocarpa* aerial parts for its phytochemical constituents, and the antimicrobial and antivirulence potential of its EO against relevant bacterial and fungal strains, to identify potential therapeutic alternatives to overcome antimicrobial resistance.

## 2. Materials and Methods

### 2.1. Plant Material

Aerial parts of *Juniperus oxycedrus* L. ssp. *macrocarpa* (Sibth & Sm.) Balland were collected by Mr. Salvatore Mura (owner of the “Fragus e Saboris de Sardigna” farm) in April 2021. Sadali (39°48′49.24″ N 9°16′25.80″ E) is a village in the province of Southern-Eastern Sardinia, in the historical sub-region of Barbagia di Seùlo. Representative plant specimens were deposited at the Herbarium S.A.S.S.A. (identified by M.U.; cumulative identification number: 16529) of the Department di Chemistry and Pharmacy, University of Sassari.

Overall, the collective quantity of plant material used for extraction was 10 kg; this was submitted to hydrodistillation in a crafted extractor (duration: 3 h); yields measured were found between 0.17–0.18% (*w*/*w*). Separation of the oils from the water was carried out via decantation, and the separated material was stored at −20 °C until analysis. The guidelines of the Italian Pharmacopeia 2008 were used to confirm EO composition and yields (using 300 g plant material and 4 h of hydrodistillation in a Clevenger-apparatus): these yields were found to be 0.19–0.20% (*w*/*w*). Drying of the oils was carried out using standard protocols (anhydrous sodium sulphate) and stored at −20 °C until analysis.

### 2.2. Oil Analyses and Quantification

Three replicates of each sample were analyzed using a Hewlett-Packard Model 5890A gas chromatography (GC) instrument, equipped with a flame ionization detector and fitted with a 60 m × 0.25 mm, thickness 0.25 μm ZB-5 fused silica capillary column (Phenomenex); relevant technical details of the GC measurements were described previously [10]. The quantification of individual compounds was expressed as an absolute weight percentage compared to using an internal standard (2,6-dimethylphenol) and response factors. GC/mass spectrometry (GC/MS) analyses were carried out with an Agilent Technologies model 7820A, connected with an MS detector 5977E MSD (Agilent), using the same conditions and column described above. Monitoring of mass units was carried out at 10–900 AMU at 70 eV, while during identification (ID), peaks between 40–900 AMU were considered. Compound ID was done to compare their retention times with those of authentic samples and/or by comparing their mass spectra with those of published data [23,24] or based on interpretation of molecular EI-fragmentation.

### 2.3. Chemicals and Reagents for Microbiological Studies

The following chemicals and culture media were used during our experiments: cation-adjusted Mueller–Hinton broth (C-MHB; Sigma-Aldrich, St. Louis, MO, USA (A), Luria–Bertani broth (LB-B; SA), Tryptic Soy broth (TSB; SA), Tryptic Soy agar (TSA; Biokar Diagnostics, Allone, Beauvais, France) RPMI-1640 (with L-glutamine and without bicarbonate, pH 7.0 with 3-(N-morpholino)-propanesulphonic acid (MOPS); Merck, Budapest, Hungary) and Dulbecco’s Modified Eagle’s Medium (DMEM; SA) were purchased. The modified Luria–Bertani agar (LB*) was prepared in-house, based on the optimized recipe described previously [1]. Ampicillin, crystal violet (CV), dimethyl sulfoxide (DMSO), 3-(4,5-dimethylthiazol-2-yl)-2,5-diphenyltetrazoliumbromide (MTT), sodium-dodecyl-sulfate (SDS), phosphate-buffered saline (PBS; pH 7.4), carbonyl cyanide 3-chlorophenylhydrazone (CCCP), ciprofloxacin, chloramphenicol, ethidium bromide (EB), erythromycin, gentamicin, kanamycin, promethazine (PMZ), reserpine and tetracycline were purchased from SA; fluconazole and XTT (2,3-bis-(2-methoxy-4-nitro-5-sulfophenyl)-2H-tetrazolium-5-carboxanilide) were purchased from Merck (Budapest, Hungary); micafungin (Molcan, Toronto, Canada), 5-fluorouracil (5-FU; Teva Pharmaceuticals; Petah Tikva, Israel [Teva]), cisplatin (Teva), doxorubicin (Teva) and thioridazine (TZ) were also purchased. For biological studies, the EO was dissolved in DMSO to obtain the relevant working concentrations, as the separation of phases was observed in cases where the volume of the broth/medium was considerably higher than the volume of the EO. Solvent concentration was always <1 *v/v%* in bacterial and fungal assays and always <2 *v/v%* in cell cytotoxicity assays; the biological effect of DMSO as a solvent—when present in these small concentrations—does not affect the results of the biological assays.

### 2.4. Bacterial Strains

The following bacterial strains were used in our experiments: *Acinetobacter baumannii* clinical isolate no. 59738 (MDR isolate), *Chromobacterium violaceum* wt85 (wild-type strain, characterized by the production of the purple violacein pigment, which is mediated by acyl-homoserine lactone (AHL) signal molecules, and capable of the production of an endogenous quorum sensing (QS) signal molecule (N-hexanoyl-L-HSL)), *C. violaceum* CV026 (Tn5 transposase-mutant, an AHL-signal molecule indicator [25]), *Clostridium perfringens* American Type Culture Collection (ATCC) 13124, *C. difficile* ATCC 9689, *Cutibacterium acnes* ATCC 11827, *Enterobacter cloacae* clinical isolate no. 31298 (isolated from wound fluid; AHL-producing-strain), *Enterococcus faecalis* ATCC 29212, *Escherhichia coli* ATCC 25922, *E. coli* K-12 AG100 (expressing the AcrAB-TolC efflux pump at its basal level), *E. coli* AG100A, an ΔAcrAB pump-deletion mutant, *Klebsiella pneumoniae* ATCC 49619, *Porphyromonas gingivalis* ATCC 33277, *Proteus mirabilis* PMI 60007, *Pseudomonas aeruginosa* PAE 170022, *P. aeruginosa* ATCC 27863 (MDR isolate), *Salmonella* Derby HWCMB 170022, *Salmonella* Enteritidis ATCC 13076, *Serratia marcescens* AS-1 (characterized by the production of an orange-red pigment prodigiosin (2-methyl-3-pentyl-6-methoxyprodigiosin) [25]), *Sphingomonas paucimobilis* Ezf 10–17 (isolated from a *Vitis vinifera* grapevine tumor; AHL-producing-strain), *Staphylococcus aureus* ATCC 25923, *S. aureus* ATCC 44300 (methicillin-resistant), *S. epidermidis* ATCC 12228, *Streptococcus pneumoniae* ATCC 49619, *S. pyogenes* ATCC 12384.

### 2.5. Fungal Isolates

Ten clinical *Candida albicans*, *C. parapsilosis sensu stricto*, *C. glabrata*, *C. tropicalis,* and *C. krusei* isolates derived from bloodstream infections were included in the study, together with the reference strains *C. albicans* SC5314, *C. parapsilosis* ATCC 22019, *C. glabrata* ATCC 90030, *C. tropicalis* ATCC 750, and *C. krusei* ATCC 6258. Furthermore, ten clinical *C. auris* isolates from three clades (South Asian, *n* = 5; East Asian, *n* = 1; South African, *n* = 1; South American, *n* = 3) were tested with the reference strain NCPF 13029 from the East-Asian clade. All isolates were identified to the species level by matrix-assisted laser desorption/ionization time-of-flight mass spectrometry (MALDI-TOF MS) [26]. In the case of *C. auris* strains, clade delineation was carried out by PCR amplification and sequencing-based on previously published methodology [27,28].

### 2.6. Determination of Minimal Inhibitory Concentrations (MIC) and Minimal Bactericidal (MBC) Concentrations on Aerobic and Facultative Anaerobic Bacterial Strains

The MIC and MBC values of the EO on the respective aerobic and facultative anaerobic strains were determined based on the recommendations of the Clinical and Laboratory Standards Institute (CLSI; M07-A11) [29]. MIC determination was performed in 96-well microtiter plates using the standard broth microdilution (BMD) method; the EO was applied at 32–0.0625 *v/v%* concentration range in the microtiter plates. After the incubation period, the MICs of the tested compounds were determined by visual inspection. During MBC determination, the dilution representing the MIC and at least two of the more concentrated EO dilutions were plated and enumerated to determine the number of viable bacteria; MBC was determined as the concentration that completely (99.9%) reduced bacterial growth when compared to the MIC dilution [29]. Where measurable MICs were >32 *v/v%*, MBCs were not determined. All experiments were performed in triplicate.

### 2.7. Determination of Antibacterial Activity against Anaerobic Bacteria Using Disk Diffusion and Broth Microdilution Methods

Screening for the antibacterial activity of the EO against anaerobic bacteria was carried out using the Kirby-Bauer disk diffusion method. Sterile filter paper disks (Whatmann MM, diameter: 6 mm) impregnated with 64 *v/v%* solutions (in 10 µL volume) of the EO were placed on Schaedler agar plates, containing 5 *v/v*% horse blood, haemin, and Vitamin K_1_ (bioMérieux, Marcy-l’Étoile, France), inoculated with the respective bacterial suspensions (with inocula of 0.5 McFarland’s standard), following the 15-15-15 rule [30]. The plates were then incubated for 48 h under anaerobic conditions in an atmosphere containing 90% N_2_, 5% H_2_, and 5% CO_2_ (Baker Ruskinn anaerobic chamber, Sanford, ME, USA). The diameters of inhibition zones produced by the EO were measured and recorded. The EO was considered inactive when the diameter of the inhibition zones was smaller than 8 mm. MIC determination of the EO against anaerobic bacteria was performed based on CLSI M11-A09 recommendations [31]. The experiments were performed in 96-well plates, using the standard BMD, and the EO was applied at 32–0.0625 *v/v%* concentration range in the microtiter plates; broth microdilution panels were prepared using Brucella broth, supplemented with 5 *v/v*% horse blood, haemin and Vitamin K_1_ (bioMérieux, Marcy-l’Étoile, France). For the inoculum, 24 h growth was used for rapid growers if sufficient growth was available from the respective strains; otherwise, 48 h growth was used. The experiments were carried out, and the 96-well plates were then incubated (for 48 h) under anaerobic conditions in an atmosphere containing 90% N_2_, 5% H_2_, and 5% CO_2_ (Baker Ruskinn anaerobic chamber, Sanford, ME, USA). All experiments were carried out in triplicate.

### 2.8. Minimal Inhibitory Concentration (MIC) Determination of Planktonic Fungal Cells

Planktonic MIC determination was performed in accordance with protocol M27-A3 of the Clinical Laboratory Standards Institute [32]. MICs of EO were determined in RPMI-1640 (with l-glutamine and without bicarbonate, pH 7.0 with MOPS; Merck, Budapest, Hungary). The tested EO concentrations ranged from 0.045 to 12.5 *v/v%*. MICs were determined as the lowest drug concentration that produces at least 50% growth reduction compared to the growth control.

### 2.9. MIC Reduction Assay for Bacteria

To test the effect of the EO on the MICs of standard antibiotics, an MIC reduction assay was performed [33]. *S. aureus* ATCC 25923, *S. aureus* ATCC 44300 (methicillin-resistant), *S. epidermidis* ATCC 12228, and *E. faecalis* ATCC 29212 were chosen as Gram-positive test microorganisms, while *E. coli* ATCC 25922, *P. mirabilis* PMI 60007, and *P. aeruginosa* PAE 170022 were included as Gram-negative test microorganisms. Ampicillin, ciprofloxacin, chloramphenicol, erythromycin, gentamicin, and tetracycline were used as reference antibiotics when relevant. The addition of the EO was carried out in fixed concentrations as adjuvants (which was determined based on the results of the MIC determination; 0.25 *v/v%* for Gram-positive and 4 *v/v%* for Gram-negative bacteria) in all the wells, except for medium control and bacterial control wells [33]. All experiments were carried out in triplicate. The MIC values of tested compounds were determined by visual inspection.

### 2.10. Assessment of Bacterial QS-Inhibitory Activity of the EO Using a Semi-Quantitative Disk Diffusion Method

The QS-inhibitory activity of the EO was performed using the disk diffusion method, as previously described [34]. Filter paper disks (Whatmann MM, diameter: 6 mm) impregnated with 10 µL of the different concentrations of the EO were placed in the center of the inoculated line(s), as described previously [34]. After the inoculation of the plates and the placement of the disks, the LB* plates were incubated for 48 h at room temperature. Assessment of the EO’s QS-inhibitory effect was carried out by measuring the diameter of the QS-inhibition zones (i.e., the size of discolored bacterial colonies (violacein or prodigiosin) with no growth inhibition). 5-fluorouracil (25 mg/mL) and thioridazine (10 mg/mL) were used as positive controls, while DMSO was used as a negative control [34]. The results of the studies are derived from the average of at least three independent experiments.

### 2.11. Bacterial Efflux Pump Inhibition Assay

The different concentrations of the EO were evaluated for their ability to inhibit efflux pumps in *E. coli* K-12 AG100 (carrying the AcrAB-TolC efflux pump, belonging to the RND (Resistance-Nodulation-Division) superfamily), *E. coli* AG100A, *S. aureus* ATCC 25923 and *S. aureus* ATCC 44300 strains (carrying the NorA efflux pump, belonging to the MFS (Major Facilitator Superfamily])group) through the real-time fluorimetry, monitoring the intracellular accumulation of ethidium bromide (EB), an efflux pump substrate [35,36]. This was determined by the automated method using a CLARIOstar Plus plate reader (BMG Labtech, Ortenberg, Germany). Reserpine (for Gram-positive bacteria) and CCCP (for Gram-negative bacteria) were applied at 25 µM as positive controls, and the solvent DMSO was applied at 1 *v/v*% as a negative control. The bacterial strains were incubated in appropriate culture media (TSB— *S. aureus* ATCC 25923 and *S. aureus* ATCC 44300; LB—*E. coli* K-12 AG100 and *E. coli* AG100A) at 37 °C until they reached an optical density (OD_600_) between 0.4 and 0.6. The culture was centrifuged at 13,000× *g* for 3 min, and the pellet was washed and resuspended with phosphate-buffered saline (PBS). The suspension was centrifuged again in the same conditions and resuspended in PBS. The EO was applied at different *v/v%* concentrations depending on their MIC values for the respective strain in a solution of a non-toxic concentration of EB (1 µg/mL) in PBS. Then, 50 µL of this solution were transferred into a 96-well black microtiter plate (Greiner Bio-One Hungary Kft, Mosonmagyaróvár, Fertősor, Hungary), and 50 µL of bacterial suspension (OD_600_ = 0.4–0.6) were added to each well. Fluorescence was measured at λ_excitation_ = 530 nm and λ_emission_ = 600 nm every minute for one hour on a real-time basis. Activity of the EO, namely the RFI of the last time point (minute 60) of the EB accumulation assay, was calculated according to the following Equation (1):(1)$$\mathrm{RFI}=\frac{{\mathrm{RF}}_{\mathrm{treated}}-{\mathrm{RF}}_{\mathrm{untreated}}}{{\mathrm{RF}}_{\mathrm{untreated}}}$$
where RF_treated_ is the relative fluorescence (RF) at the last time point of the EB accumulation curve in the presence of the compound, and RF_untreated_ is the RF at the last time point of the EB accumulation curve of the untreated control, having only the solvent (DMSO) control [35,36]. The samples were tested in triplicate, and the RFI values presented come from the average of these three values.

### 2.12. Inhibition of Bacterial Biofilm-Formation

Biofilm-forming ability of *S. aureus* ATCC 25923 and *S. aureus* ATCC 44300 strains was studied in 96-well microtiter plates, using tryptic soy broth (TSB) in the presence of the EO, as previously described [35]. Compounds were added individually, starting at 1/2 MIC (1 *v/v%* to 0.0625 *v/v%* for *S. aureus* ATCC 25923, 32 *v/v%* to 2 *v/v%* for *S. aureus* ATCC 44300). PMZ was applied at 25 µM as a positive control, and the solvent DMSO was applied at 1 *v/v*% as a negative control. Biofilm formation was determined by measuring the OD at 600 nm using a FLUOstar Optima plate reader (BMG Labtech, Aylesbury, UK) [8]. The anti-biofilm effect of the EO was expressed in the percentage (%) decrease in biofilm formation [35]. The assay was repeated a minimum of three times.

### 2.13. Antifungal Susceptibilty Testing of Biofilms

Biofilm forming ability in fungi was evaluated with the CV assay as previously described by O’Toole [37]; biofilm development was considered if the OD value at 540 nm was higher than 0.15. *Candida* isolates were suspended in RPMI-1640 broth in concentrations of 1 × 10^6^ cells/mL, and aliquots of 100 µL were inoculated onto flat-bottom 96-well sterile microtitre plates (TPP, Trasadingen, Switzerland) and then incubated statically at 37 °C for 24 h to produce one-day-old biofilms [38,39]. The examined EO concentrations for sessile MIC determination ranged from 0.045 to 12.5 *v/v%*. The biofilms were washed three times with sterile physiological saline. Afterward, MIC determination was performed in RPMI-1640 using XTT-assay. The percentage change in metabolic activity was calculated based on absorbance (*A*) at 492 nm as 100% × (*A*_well_ − *A*_background_)/(*A*_drug-free well_ − *A*_background_). MICs of biofilms were defined as the lowest drug concentration resulting in at least 50% metabolic activity reduction compared to control wells [38,39].

### 2.14. Evaluation of Interactions by Fractional Concentration Index (FICI)

Interactions between tested antifungals (fluconazole and micafungin) and EO were assessed using two-dimensional broth microdilution chequerboard assay. Afterwards, interactions were analyzed using FICI determination [38,39,40]. The tested concentration range of EO was the same as described above for planktonic and biofilm MIC determination. The tested fluconazole (Merck, Budapest, Hungary) concentrations ranged from 2 mg/L to 128 mg/L, 8 mg/L to 512 mg/L, and 0.5 mg/L to 32 mg/L for *C. auris* planktonic cells, *C. auris* sessile cells, and *C. albicans* biofilms, respectively. Micafungin (Molcan, Toronto, ON, Canada) concentrations ranged from 4 mg/L to 256 mg/L and from 0.015 mg/L to 1 mg/L for *C. auris* and *C. albicans* biofilms, respectively. FICIs were calculated using the following formula: ΣFIC = FIC_A_ + FIC_B_ = MIC_A_^comb^/MIC_A_^alone^ + MIC_B_^comb^/MIC_B_^alone^, where MIC_A_^alone^ and MIC_B_^alone^ stand for MICs of drugs A and B when used alone, and MIC_A_^comb^ and MIC_B_^comb^ represent the MIC values of drugs A and B in combination at isoeffective combinations, respectively. FICI was determined as the lowest ΣFIC [38,40]. If the obtained MIC value is higher than the highest tested drug concentration, the next highest twofold concentration was considered MIC. FICI values of ≤0.5 were defined as synergistic, between >0.5 and 4 as indifferent, and >4 as antagonistic.

### 2.15. Cell Culture

The NIH/3T3 (ATCC CRL-1658) mouse embryonic fibroblast cell line (LGC Promochem) was cultured in Dulbecco’s Modified Eagle’s Medium (DMEM with 4.5 g/L glucose), supplemented with 10% heat-inactivated fetal bovine serum (FBS), 2 mM L-glutamine, 1 mM Na-pyruvate, nystatin, and a penicillin–streptomycin mixture in concentrations of 100 U/L and 10 mg/L, respectively. The cell lines were incubated in a humidified atmosphere (5% CO_2_, 95% air) at 37 °C.

### 2.16. Assay for Cytotoxic Effect

The effects of the EO on cell growth were tested on the NIH/3T3 mouse embryonic fibroblast cell line. The adherent mouse embryonic fibroblast cell line (seeded at 10^4^/well cell density in the 96-well microtiter plates 4 h before the assay) were treated with the EO in dilutions starting from 32 *v/v%*. Cisplatin, 5-FU, and doxorubicin were used as positive controls, while DMSO was used as solvent control. The protocol for an MTT (thiazolyl blue tetrazolium bromide)-based cell viability assay was described previously [41]. Cell growth was determined by measuring the optical density (OD) at 540 nm (ref. 630 nm) with a Multiscan EX ELISA reader (Thermo Labsystems, Cheshire, WA, USA), and the percentage of inhibition of cell growth was determined according to the following Equation (2) [42]:(2)$${\mathrm{IC}}_{50}=100-\left[\frac{OD\;sample-OD\;\;medium\;control}{OD\;cell\;control-OD\;medium\;control}\right]\times 100$$

IC_50_ values and the SD of the triplicate experiments were calculated using GraphPad Prism software version 5.00 for Windows (GraphPad Software, San Diego, CA, USA; available at www.graphpad.com, accessed on 23 February 2022).

## 3. Results

### 3.1. Chemical Composition of EO from Juniperus oxycedrus *L. ssp.* macrocarpa

The detector response factors (RFs) were determined for the key components relative to 2,6-dimethylphenol and assigned to other components based on the functional group and/or structural similarity since oxygenated compounds have lower detectability by flame ionization detector (FID) than hydrocarbons. The standards were >95% also, and actual purity was checked by GC. Several response factor solutions were prepared that consisted of only four or five components (plus 2,6-dimethylphenol) to prevent interference from trace impurities. It is known that the oxygenated compounds have lower sensitivity than the hydrocarbons to FID. We calculated the response factor using a standard mixture of α-pinene, α-terpineol, neral, geranial, geranyl acetate, and caryophyllene; in this mixture, terpenes accounted for 92% of the mixture, aldehydes ~5% and alcohols, esters and sesquiterpenes ~1% each. In our analyses, we obtained that the RF of hydrocarbons was equal to 1 while for alcohols it was 0.80 and for esters 0.71. For this reason, we have multiplied the experimental data obtained for the following correction factors: 1 for hydrocarbons, 1.24 for aldehydes and ketones, 1.28 for alcohols and 1.408 for esters. The most present organic chemical compounds are: α-pinene (56.63% ± 0.24); limonene (14.66% ± 0.11); β-pinene (13.42% ± 0.09) (Table 1).

### 3.2. Antibacterial Activity of the EO

The antibacterial activity of the EO was tested against a variety of Gram-positive and Gram-negative bacteria in vitro. Based on our results, the EO presented with more potent antibacterial properties against common Gram-positive strains (MIC range: 0.25–2 *v/v%*), while the EO’s potency was much lower in respect to Gram-negative bacteria (from 16 to over 32 *v/v%*, with the exception of *Salmonella*). Similar results were observed for anaerobic bacteria; however, the EO possessed no antimicrobial effects on *C. difficile*, even though the strain is Gram-positive. The EO had no effect on the MDR strains (MRSA, MDR *Acinetobacter,* and *Pseudomonas*) involved in our experiments (Table 2 and Table 3).

### 3.3. MIC Reduction Assay

In our MIC reduction assays, the EO was applied in fixed concentrations to ascertain whether it possessed MIC-modulating properties for commonly used antibiotics. While the EO showed MIC-reducing properties in some cases for tetracycline, erythromycin/gentamicin, and chloramphenicol, the most pronounced activity was seen for the ciprofloxacin-EO combination, where MIC values were 4–8-times lower, compared to the native antimicrobial activity of the antibiotic. On the other hand, the effect of ampicillin was not enhanced in any form due to the EO treatment (Table 4 and Table 5).

### 3.4. Efflux Pump Inhibition

A real-time EB-bromide accumulation assay was performed to determine whether EO possesses the potency to inhibit various bacterial efflux pumps, which are of pivotal importance in developing the MDR phenotype. To ensure that the fluorescence of the EO itself did not influence our measurements, a control experiment was performed where the EO was tested alone in PBS against an EB solution and a solution of EB and the compound together. Based on the real-time fluorimetry measurements, the EO increased the measured fluorescence levels—compared to levels observed after the treatment with the positive controls (CCCP for *E. coli* AG100 and AG100A, and reserpine for *S. aureus* ATCC 25923 and ATCC 44300), which may be a direct correlate of the inhibition of EB-efflux from the tested bacteria. For *E. coli* AG100 and AG100A, the EO at 8 *v/v%* and 4 *v/v%* exerted efflux pump-inhibitory activity 98.4% and 71.5% higher, and 242.9% and 216.8% higher than CCCP, respectively. The EO also showed pronounced activity in the same concentrations in the case of MRSA (fluorescence measurements were 290.9% and 233.3% higher than reserpine), and in lower concentrations (adjusted due to the differences in MIC) for ATCC 25923, with fluorescence reads 188.6% and 187.7% higher than reserpine, in 4 *v/v%* and 2 *v/v%* (Table 6).

### 3.5. Inhibition of Biofilm-Formation and QS in Bacteria, Cytotoxicity of Fibrolast Cells

Unlike the positive controls (TZ, 5-FU), the EO did not have any QS-inhibitory effects on any of our tested model organisms (*S. marcescens* AS-1, *C. violaceum* CV wt85, *C. violaceum* CV 026 + *S. paucimobilis* Ezf 10–16, *C. violaceum* CV 026 + *E. cloacae* CI 31298) in the relevant concentration-range (EO 32–0.0625 *v/v%*), i.e., they did not affect the signal molecule-mediated pigment (violacein or prodigiosin) production of these bacteria in in vitro conditions, thus not showing a QS “inhibition zone”, similar to TZ and 5-FU. Likewise, the EO showed no potency to significantly inhibit the biofilm-production of *S. aureus* ATCC 25923 and *S. aureus* ATCC 44300 (with inhibition percentages in the ~0–5% range) in the relevant concentration range, in comparison to PMZ (33.32 ± 2.31% and 41.78 ± 1.35 %, respectively) (Table 7). The EO did not have cytotoxic properties on the tested mouse embryonic fibroblast cell (NIH/3T3) lines up to >32 *v/v%* (Table 8).

### 3.6. Antifungal Activity, Combination (FICI) Assay

The median and range of the MIC values to EO for planktonic and sessile cells of *Candida* isolates are shown in Table 8 and Table 9. The median planktonic MICs observed for the tested isolates showed 8-fold, 32-fold, 16-fold, 4-fold, 4-fold increases for *C. albicans*, *C. parapsilosis*, *C. glabrata*, *C. tropicalis,* and *C. krusei*, respectively. It is noteworthy that, a 64-fold MIC increase was observed in the case of *C. auris* median MIC values for biofilms compared to planktonic cells (Table 9 and Table 10).

Two species were chosen for in vitro combination-based experiments. Table 11 summarizes the in vitro interactions between micafungin, fluconazole, and the EO, based on calculated FICI values against *C. albicans* and *C. auris*. An antagonistic interaction was never observed (all FICIs ≤ 4). The EO exerted a synergistic interaction with fluconazole both against *C. albicans* (median FICI was 0.139) and *C. auris* sessile cells (median FICI was 0.278), while the planktonic interaction was indifferent (median FICI was 1). The EO synergistically enhanced the activity of micafungin against *C. auris* sessile cells (median FICI was 0.088) but not against *C. albicans* biofilms (median FICI was 1).

## 4. Discussion

AMR is one of the most daunting issues facing humanity in the 21st century, as drug-resistant pathogens lead to longer hospital stays (affecting economic turnover), worse treatment outcomes, and excess mortality [42]. Among bacterial infections, the so-called “ESKAPE” pathogens have raised the most concerns based on their prevalence and overall mortality [43]. Global and intersectoral efforts need to be made to address the daunting issue of AMR; one of these efforts is the development of novel antimicrobial drugs to increase the pool of clinically-available drugs [44,45]. In addition to directly-acting (static and/or cidal) antimicrobial agents, it has been suggested that the development of antimicrobial adjuvants (i.e., resistance-reversing agents), anti-biofilm, and anti-virulence compounds are also viable strategies to address AMR in a clinical environment, as they (a) allow for the use of already existing drugs and (b) do not lead to strong selection pressure, leading to the rapid development of resistant clones [46,47]. While the richness of the “chemical space” of compounds found from natural sources is well-known, there has been a peak in interest in the isolation and characterization of plant extracts and secondary metabolites as potential antimicrobial agents (partly due to the availability of more precise technologies in chemistry) [48]. Although there are still gaps in our knowledge, EOs have been widely regarded as one of the most clinically-relevant compounds of natural origin, owing to their diverse chemical composition and numerous potential applications [49]. In fact, given the extensive negative environmental impacts of antimicrobials—per the One Health paradigm—the use of environment-friendly alternatives, such as EOs, puts these studies in an additional context [50]. EOs are usually characterized by a complex chemical composition, which may be further influenced by the part of the plant where the EO is stored, environmental factors, and isolation conditions [51]. With their complex composition, the constituents of EOs may have multifaceted interactions with each other; even trace elements, present in very low concentrations, may impact their biological effects [52]. In addition to this, novel formulation technologies—e.g., micellar EOs, liposomes, nanocarriers—may further enhance the bioavailability of these compounds [53,54,55].

Despite the wide-ranging efforts to characterize the *Juniperus* genus, their therapeutic potential has not been fully characterized. In our current efforts, the phytochemical characterization and in vitro antimicrobial analysis of the EO originating from the aerial parts of *Juniperus oxycedrus* L. ssp. *macrocarpa* was performed, which is endemic to the Sardinian Island. The EO was isolated from the plant material with a yield of ~0.2 *w*/*w*%, of which the major constituents were α-pinene (>50%), β-pinene (>10%), and limonene (>10%). Our results have shown that the EO, which was proven to be non-toxic on fibroblast cells in the tested concentration range, had potent antibacterial activity against common Gram-positive bacteria—both aerobes and anaerobes—while it had no effect on the methicillin-resistant counterpart of *S. aureus*. Additionally, the EO was considerably (8–64-times) less effective against most of the Gram-negative bacteria tested; notable exceptions from these patterns include the non-susceptibility of *C. difficile* to the EO and the susceptibility of *Salmonella* species tested. *C. difficile*—one of the most common nosocomial pathogens leading to substantial morbidity and mortality—showed non-susceptibility to a wide range of antibiotics and antimicrobial agents, owing to its intrinsic resistance determinants, resistance genes acquired via horizontal gene transfer (HGT) and due to its unique physiology; the sum of these factors lead to difficulties in the treatment of these infections [56]. On the other hand, the serendipitous susceptibility of *Salmonella* species against single and blended EOs has already been reported, with α-pinene, α-terpineol, carvacrol, and eugenol having a principal role in their anti-*Salmonella* effects (having MICs in the 0.05–0.5 mg/L range) [57]. In fact, the use of such EOs has been widely proposed as biocontrol agents to eliminate the spread of these foodborne in the food industry [58].

Mechanistic studies are needed to understand the complex mechanism of antibacterial action of the EO isolated from *Juniperus oxycedrus* L. ssp. *macrocarpa,* although in many cases, EOs act not by one single mechanism but through several pathways to varying extents; nevertheless, our results showed that the cell wall composition of bacteria (Gram-positive or negative) was a critical determining factor in the efficacy of the EO, implying that the permeability of the cell wall to the EO may, in fact, modulate its antibacterial effectiveness. This was further underlined by the fact that this trend was also seen in strict anaerobes, suggesting that the mechanism of action is likely not related to the microbial metabolism/respiration of the cells (which would be similar in the case of all anaerobes) [59]. The preferential action of EOs towards Gram-positive bacteria has been documented extensively in the literature: recently, during the study of EOs from the leaves of *Paramignya trimera* (Oliv.) Guillaum and *Limnocitrus littoralis* (Miq.) Swingle, Le et al. [60], from the leaves of *Leoheo domatiophorus* Chaowasku, D.T. Ngo and H.T. Le by Le et al. [61], in the study of EOs from the leaves, rhizomes, and whole plant of *Hornstedtia bella* Škornik by Donadu et al. [62], during the study of the EO from the weed *Austroeupatorium inulaefolium* by Bua et al. [63], and the study of water extracts of *Borojoa patinoi* Cuatrecasas by Chaves-López et al. [64]. The variation in the efficacy of EOs to exert their antibacterial activities on bacteria may (at least, in part) be explained by the differences in the constitution of the cell walls. The thick, largely hydrophobic peptidoglycan cell wall allows for greater penetration of EOs towards the cell membrane and the intracellular space. The EO components lead to disruption in the cell wall and lipid bilayer, leading to the disarray of metabolic processes and cell lysis [65]. On the other hand, the cell wall of Gram-negative bacteria is more complex, with characteristic hydrophobic (a much less pronounced peptidoglycan fraction) and hydrophilic components (e.g., porin channels, lipopolysaccharide); this makes the penetration of the hydrophobic EOs towards the inner membrane slow and cumbersome, which often leads to less pronounced antibacterial effect [66,67].

The EO presented potent efflux pump inhibitory effects against transport proteins from two major superfamilies (MFS: *S. aureus*, RND: *E. coli*). This may explain the results seen in the MIC reduction assay; in the latter experiment, the EO was applied as an adjuvant to increase the efficacy of reference antibiotics, showing that the MICs of ciprofloxacin and tetracycline were considerably decreased in the case of many Gram-positive and Gram-negative bacterial strains. Interestingly, fluoroquinolones and tetracycline-derivatives are commonly associated with moderate-to-high-level resistance caused by the overexpression of bacterial efflux pumps. At the same time, the treatment of the bacteria with the EO may have led to the inhibition of said pumps, leading to better antibacterial effectiveness [68]. The fact that similar significant decreased MICs were not seen with other protein synthesis inhibitor drugs (which have a similar target as tetracycline) further validates this hypothesis.

Regarding the *Candida* species, the highest activity of EO was observed in the case of *C. auris* isolates both against planktonic cells and one-day-old biofilms. *C. auris* poses a global health threat due to its ability to cause clonal nosocomial outbreaks with a high mortality rate [69]. Moreover, a high percentage of clinical isolates show a resistant phenotype to one or two major classes of traditional antifungal drugs, especially in the case of azoles and polyenes [70]. Based on current therapeutic recommendations, echinocandins are the first-line drugs for treating *C. auris* infections; however, the number of echinocandins and pan-resistant isolates is steadily increasing, especially in the USA [70]. Similar to antibacterial agents, antifungal drug discovery is a slow and challenging process, particularly for newly emerged fungal species such as *C. auris*. Nevertheless, the development of new antifungal therapeutic approaches has high priority. To date, the number of research investigating the effect of EOs and their components against *C. auris* planktonic cells and biofilms is limited. In this study, the *Juniperus*-derived EO showed surprisingly high activity against *C. auris* planktonic cells and biofilms compared to the susceptibility of other clinically relevant *Candida* species tested. Tran et al. investigated the in vitro antifungal activity of *Cinnamomun zeylanicum* bark and leaf essential oils against *C. auris* [71]. Essential oils derived from the *C. zeylanicum* Blume bark and leaves showed a potent fungicidal effect from 0.03% (*v*/*v*) [71]. In addition, EOs significantly inhibited hemolysin production and altered the fungal morphology. Presumably, the observed antifungal effect can be explained by the membrane damage exerted by *C. zeylanicum* [54]. De Alteriis et al. tested the free and encapsulated form of EO from *Lavandula angustifolia* [54]. They found that the tested compound could eradicate primary and persister-derived sessile communities of *C. auris*. Based on their findings, the antifungal effect may be explained by reactive oxygen species production and alteration of expression of biofilm-related genes [72]. Our study had a limitation, namely the shortage of investigation of antifungal effect-related molecular mechanisms exerted by EO against *C. auris*. Presumably, the main mechanism is based on its lipophilic property, which enhances the penetration of hydrophobic compounds to the cytoplasmic membrane leading to membrane damage [72,73]. Moreover, the active component(s) of oil may generate an extensive reactive oxygen species production, too [72,73]. In this paper, we performed the characterization and the in vitro antimicrobial activity assessment of endemic *Juniper* EOs to understand where to orient our future studies for the technological-pharmaceutical formulation of this essential oil. Based on our results, the EO shows strong activity aimed at *Candida* spp. (both planktonic and biofilm-embedded) and Gram-positive bacteria. In addition, the EO has potential efflux pump inhibitory properties. Thus, in the future we aim to include formulating—in collaboration with other biomedical engineering colleagues—biodegradable membranes based on lipophilic polymers with our EO. The latter applied to chronically infected wounds (e.g., for diabetic patients who require anti-microbial and anti-biofilm substances) could help in speeding up healing processes with an innovative method, which has a natural origin and that can be practically performed effectively also in terms of production from an industrial point of view.

## Figures and Tables

**Table 1 microorganisms-10-00758-t001:** Chemical composition of the EO from the aerial parts of *Juniperus oxycedrus* L. ssp. *macrocarpa*, based on GC/MS analyses (using a no-polar column ZB-5).

Rt	RI Apol Lett	RI Apol Sper	Constituents	Percentage (%)	ID *
10.38	700	704	heptane	0.05 ± 0.01	RI, MS
21.50	920	920	β-thujene	0.10 ± 0.02	Std
22.08	937	939	α-pinene	56.63 ± 0.24	Std
22.89	945	953	α-fenchene	0.05 ± 0.01	Std
23.01	956	955	camphene	1.50 ± 0.02	Std
23.59	970	974	benzaldehyde	0.04 ± 0.01	RI, MS
24.14	975	977	α-sabinene	0.33 ± 0.03	Std
24.59	979	981	β-pinene	13.42 ± 0.09	Std
24.90	991	992	β-myrcene	0.73 ± 0.04	Std
25.94	1004	1003	pseudolimonene	0.03 ± 0.01	RI, MS
26.06	1003	1005	α-phellandrene	0.04 ± 0.01	Std
26.22	1008	1011	δ-3-carene	0.66 ± 0.02	Std
26.64	1017	1015	α-terpinene	0.30 ± 0.02	Std
27.06	1025	1026	p-cymene	0.51 ± 0.03	Std
27.36	1029	1027	limonene	14.66 ± 0.11	Std
27.46	1026	1030	benzyl alcohol	3.41 ± 0.05	RI, MS
27.59	1026	1031	1,8-cineole	1.37 ± 0.03	Std
28.88	1060	1064	γ-terpinene	0.16 ± 0.02	Std
30.39	1088	1087	α-terpinolene	0.14 ± 0.01	Std
32.46	1129	1128	*cis*-allo-ocimene	3.00 ± 0.07	RI, MS
33.16	1131	1131	*trans*-allo-ocimene	0.25 ± 0.03	RI, MS
33.59	1137	1141	*trans*-sabinol	0.06 ± 0.01	RI, MS
35.17	1169	1166	endo-borneol	0.03 ± 0.01	RI, MS
35.50	1177	1180	terpinen-4-ol	0.13 ± 0.02	Std
35.67	1179	1183	p-cymen-8-ol	0.02 ± 0.01	RI, MS
36.13	1186	1180	α-terpineol	0.08 ± 0.02	Std
39.99	1189	1287	bornyl acetate	0.08 ± 0.01	RI, MS
42.39	1350	1352	α-cubebene	0.03 ± 0.01	Std
42.73	1350	1353	α-longipinene	0.02 ± 0.01	RI, MS
45.06	1419	1419	β-caryophyllene	0.40 ± 0.08	Std
46.17	1452	1454	humulene	0.07 ± 0.02	Std
46.58	1480	1480	γ-muurolene	0.03 ± 0.01	RI, MS
46.89	1485	1482	germacrene D	0.05 ± 0.01	Std
47.78	1521	1523	δ-cadinene	0.10 ± 0.02	Std
47.93	1529	1530	calamenene	0.02 ± 0.01	RI, MS
49.81	1582	1583	caryophyllene oxide	0.07 ± 0.02	Std
			**Total**	**99.72**	

* Rt: retention time; RI: identification by comparison of retention index values with those reported in literature; Std: identification by comparison of the retention time and mass spectrum of available authentic standards; MS: identification by comparison of the MS databases (Adams, Nist) and by interpretation of the MS fragmentations.

**Table 2 microorganisms-10-00758-t002:** Antibacterial activity of the EO from the aerial parts of *Juniperus oxycedrus* L. ssp. *macrocarpa* on Gram-positive aerobic and anaerobic bacteria.

	MIC	MBC
*Enterococcus faecalis* ATCC 29212	**1 *v/v%***	**2 *v/v%***
*Staphylococcus aureus* ATCC 25923	**2 *v/v%***	**4 *v/v%***
*S. aureus* ATCC 44300 (MRSA)	>32 *v/v%*	−
*S. epidermidis* ATCC 12228	**1 *v/v%***	**2 *v/v%***
*Streptococcus pneumoniae* ATCC 49619	**0.25 *v/v%***	**0.5 *v/v%***
*S. pyogenes* ATCC 12384	**0.25 *v/v%***	**0.5 *v/v%***
*Clostridium perfringens* ATCC 13124	Disk diffusion diameter: 12 mm	MIC: **8 *v/v%***
*C. difficile* ATCC 9689	Disk diffusion diameter: 0 mm	MIC: >32 *v/v%*
*Cutibacterium acnes* ATCC 11827	Disk diffusion diameter: 19 mm	MIC: **4 *v/v%***

MIC: minimum inhibitory concentration; MBC: minimum bactericidal concentration; Values in **bold** represent measurable antibacterial activity.

**Table 3 microorganisms-10-00758-t003:** Antibacterial activity of the EO from the aerial parts of *Juniperus oxycedrus* L. ssp. *macrocarpa* on Gram-negative aerobic and anaerobic bacteria.

	MIC	MBC
*Acinetobacter baumannii* CI 59738 (MDR)	>32 *v/v%*	-
*Chromobacterium violaceum* wt85	**16 *v/v%***	>64 *v/v%*
*C. violaceum* CV026	**16 *v/v%***	>64 *v/v%*
*Enterobacter cloacae* CI 31298	**16 *v/v%***	>64 *v/v%*
*E. coli* ATCC 25922	**16 *v/v%***	>64 *v/v%*
*Escherichia coli* K-12 AG100	**16 *v/v%***	-
*E. coli* AG100A	**16 *v/v%***	>64 *v/v%*
*Klebsiella pneumoniae* ATCC 49619	**16 *v/v%***	>64 *v/v%*
*Proteus mirabilis* PMI 60007	**16 *v/v%***	>64 *v/v%*
*Pseudomonas aeruginosa* PAE 170022	>32 *v/v%*	-
*P. aeruginosa* ATCC 27863	>32 *v/v%*	-
*Salmonella* Derby HWCMB 170022	**4 *v/v%***	16 *v/v%*
*Salmonella* Enteritidis ATCC 13076	**4 *v/v%***	8 *v/v%*
*Serratia marcescens* AS-1	**16 *v/v%***	>64 *v/v%*
*Sphyngomonas paucimobilis* Ezf 10–17	>32 *v/v%*	-
*Bacteroides fragilis* ATCC 25285	Disk diffusion diameter: 0 mm	MIC: >32 *v/v%*
*Porphyromonas gingivalis* ATCC 33277	Disk diffusion diameter: 0 mm	MIC: >32 *v/v%*

CI: clinical isolate; MDR: multidrug resistant; MIC: minimum inhibitory concentration; MBC: minimum bactericidal concentration; Values in **bold** represent measurable antibacterial activity.

**Table 4 microorganisms-10-00758-t004:** Results of MIC reduction assay using reference antibiotics and the EO from the aerial parts of *Juniperus oxycedrus* L. ssp. *macrocarpa* on Gram-positive bacteria.

	Ampicillin		Ciprofloxacin		Tetracycline		Erythromycin		Chloramphenicol	
Treatment: 0.25 *v/v%* EO	UT (mg/L)	T (mg/L)	UT (mg/L)	T (mg/L)	UT (mg/L)	T (mg/L)	UT (mg/L)	T (mg/L)	UT (mg/L)	T (mg/L)
*S. aureus* ATCC 25923	0.25	0.25	0.125	**0.0312**	0.5	**0.0625**	0.5	0.5	2	**1**
*S. aureus* ATCC 44300 (MRSA)	>128	>128	16	**8**	32	32	>128	>128	2	**1**
*S. epidermidis* ATCC 12228	0.125	0.125	0.125	**0.0156**	4	**1**	4	**2**	1	**0.25**
*E. faecalis* ATCC 29212	4	**2**	0.25	**0.0625**	>128	>128	8	8	>128	>128

T: treated; UT: untreated; Values in **bold** represent decreased MIC values due to treatment with the EO.

**Table 5 microorganisms-10-00758-t005:** Results of the MIC reduction assay using reference antibiotics and the EO from the aerial parts of *Juniperus* oxycedrus L. ssp. *macrocarpa* on Gram-negative bacteria.

	Ampicillin		Ciprofloxacin		Tetracycline		Gentamicin		Chloramphenicol	
Treatment: 4 *v/v%* EO	U	T	UT	T	UT	T	UT	T	UT	T
	(mg/L)	(mg/L)	(mg/L)	(mg/L)	(mg/L)	(mg/L)	(mg/L)	(mg/L)	(mg/L)	(mg/L)
*E. coli* ATCC 25922	2	**1**	0.125	**0.0156**	0.125	**0.0625**	0.25	**0.0625**	2	2
*K. pneumoniae* ATCC 49619	32	32	0.25	**0.0625**	0.125	**0.0625**	0.5	**0.25**	8	**4**
*P. mirabilis* PMI 60007	>128	>128	0.5	**0.0625**	>128	>128	64	64	64	64
*P. aeruginosa* PAE 170022	>128	>128	0.5	**0.125**	64	64	0.25	0.25	32	32

Values in **bold** represent decreased MIC values due to treatment with the EO.

**Table 6 microorganisms-10-00758-t006:** Relative fluorescence index (RFI) values for the tested bacterial strains after treatment with the EO from the aerial parts of *Juniperus oxycedrus* L. ssp. *macrocarpa*.

	RFI ± SD			
Compounds	*E. coli* K-12 AG100	*E. coli* AG100A	*S. aureus* ATCC 25923	*S. aureus* ATCC 44300
EO 32 *v/v%*	-	-	-	−0.42 ± 0.05 ^4^
EO 16 *v/v%*	-	-	-	0.06 ± 0.03 ^4^
EO 8 *v/v%*	**3.83 ± 0.29** ^1^	**2.89 ± 0.19** ^2^	-	**0.96 ± 0.09** ^4^
EO 4 *v/v%*	**3.31 ± 0.19** ^1^	**2.58 ± 0.20** ^2^	**2.15 ± 0.16** ^3^	**0.77 ± 0.10** ^4^
EO 2 *v/v%*	**3.06 ± 0.35** ^1^	**1.98 ± 0.13** ^2^	**2.14 ± 0.15** ^3^	**0.65 ± 0.03** ^4^
EO 1 *v/v%*	1.65 ± 0.14 ^1^	**1.21 ± 0.11** ^2^	**1.28 ± 0.08** ^3^	-
EO 0.5 *v/v%*	0.92 ± 0.10 ^1^	1.03 ± 0.05 ^2^	1.07 ± 0.10 ^3^	-
EO 0.25 *v/v%*	-	-	0.66 ± 0.03 ^3^	-
Reserpine (25 µM)	-	-	*1.14 ± 0.09* ^3^	*0.33 ± 0.06* ^4^
CCCP (25 µM)	*1.93 ± 0.02* ^1^	*1.19 ± 0.12* ^2^	-	-

^1–4^ The value of the positive control in each different assay; superscript numbers are relative to the positive control obtained in each assay. SD: standard deviation. Values in **bold** show higher RFI values compared to the positive control.

**Table 7 microorganisms-10-00758-t007:** Biofilm and quorum sensing inhibitory activity of the EO from the aerial parts of *Juniperus oxycedrus* L. ssp. *macrocarpa*.

EO	Biofilm Inhibition (%) ± SD		Quorum Sensing Inhibition (mm) ± SD			
	*S. aureus* ATCC 25923	*S. aureus* ATCC 44300	*S. marcescens* AS-1	*C. violaceum* CV wt85	*C. violaceum* CV 026 + *S. paucimobilis* Ezf 10–16	*C. violaceum* CV 026 + *E. cloacae* CI 31298
32 *v/v%*	−	4.86 ± 2.01%	−	−	−	−
16 *v/v%*	−	0	−	−	−	−
8 *v/v%*	−	0	0	0	0	0
4 *v/v%*	−	0	0	0	0	0
2 *v/v%*	−	0	0	0	0	0
1 *v/v%*	1.25 ± 1.34%	−	0	0	0	0
0.5 *v/v%*	0.37 ± 0.45%	−	−	−	−	−
0.25 *v/v%*	0	−	−	−	−	−
0.125 *v/v%*	0	−	−	−	−	−
0.0625 *v/v%*	0	−	−	−	−	−
DMSO (1 *v/v%*)	0	0	0	0	0	0
PMZ (25 µM)	33.32 ± 2.31%	41.78 ± 1.35%	NR	NR	NR	NR
TZ (10 mg/mL)	NR	NR	23.7 ± 2.3	21.0 ± 1.2	21.2 ± 2.0	19.6 ± 1.6
5-FU (25 mg/mL)	NR	NR	46.2 ± 3.1	41.2 ± 2.8	36.3 ± 1.2	29.8 ± 1.4

NR: not relevant; SD: standard deviation; 0: no activity.

**Table 8 microorganisms-10-00758-t008:** Cytotoxic activity of the EO from the aerial parts of *Juniperus oxycedrus* L. ssp. *macrocarpa* and various anticancer drugs on mouse embryonic fibroblast cells (NIH/3T3).

Compounds	IC_50_ (µM or *v/v%*) ± SD
**Cisplatin (+)**	11.16 ± 0.08
**Doxorubicin (+)**	13.58 ± 0.14
**5-FU (+)**	38.72 ± 1.94
**EO**	>32 *v/v%*
**DMSO (−)**	>2 *v/v%*

(+): positive control, (−): negative control.

**Table 9 microorganisms-10-00758-t009:** Minimum inhibitory concentration (MIC) of the EO from the aerial parts of *Juniperus oxycedrus* L. ssp. *macrocarpa* against reference strains of *Candida albicans*, *C. parapsilosis*, *C. glabrata*, *C. tropicalis*, *C. krusei* and *C. auris* planktonic cells and one-day-old biofilms.

Reference Strains	Planktonic MIC Values	Sessile MIC Values
	(*v/v%*)	(*v/v%*)
*Candida albicans* SC 5314	0.39	3.12
*Candida parapsilosis* ATCC 22019	0.39	No biofilm production
*Candida glabrata* ATCC 90030	0.39	0.78
*Candida tropicalis* ATCC 750	1.56	No biofilm production
*Candida krusei* ATCC 6258	0.09	No biofilm production
*Candida auris* NCPF 13029	0.02	No biofilm production

**Table 10 microorganisms-10-00758-t010:** Minimum inhibitory concentration (MIC) of the EO from the aerial parts of *Juniperus oxycedrus* L. ssp. *macrocarpa* against clinical *Candida* isolates.

Species	Planktonic MIC Values	Sessile MIC Values
(Number of Isolates)	Median (Range) (*v/v%*)	Median (Range) (*v/v%*)
*Candida albicans* (*n* = 10)	0.78 (0.39–0.78)	6.25 (0.78–12.5)
*Candida parapsilosis* (*n* = 10)	0.19 (0.09–0.39)	6.25
*Candida glabrata* (*n* = 10)	0.19 (0.02–0.39)	3.12 (3.12–6.25)
*Candida tropicalis* (*n* = 10)	3.12 (1.56–6.25)	12.5 (6.25–12.5)
*Candida krusei* (*n* = 10)	1.56 (0.78–3.125)	6.25
*Candida auris* (*n* = 10)	0.02 (0.02–0.04)	1.56 (0.78–12.5)

**Table 11 microorganisms-10-00758-t011:** Minimum inhibitory concentration (MIC) of fluconazole, micafungin alone, and in combination with the EO from the aerial parts of *Juniperus oxycedrus* L. ssp. *macrocarpa*, against *Candida albicans* and *Candida auris* planktonic cells and one-day-old biofilms. Furthermore, in vitro interactions by fractional inhibitory concentration index (FICI) of fluconazole and micafungin in combination with EO against planktonic cells and biofilms.

Species (Number of Isolates)	Median MIC (Range) of Drug Used				FICI Median (Range)	Nature of Interaction
	Alone		In Combination			
**Planktonic *C. albicans* (*n* = 5)**	**Fluconazole mg/L**	**EO *v/v%***	**Fluconazole mg/L**	**EO *v/v%***	N.D.	
	N.D.	N.D.	N.D.	N.D.		
	Micafungin	EO *v/v%*	Micafungin	EO *v/v%*	N.D.	
	mg/L		mg/L			
	N.D.	N.D.	N.D.	N.D.		
**Planktonic *C. auris* (*n* = 5)**	Fluconazole	EO *v/v%*	Fluconazole	EO *v/v%*	1	Indifferent
	mg/L		mg/L			
	32 (32–>32)	0.02 (0.02–0.04)	16	0.01		
	Micafungin	EO *v/v%*	Micafungin	EO *v/v%*	N.D.	
	mg/L		mg/L			
	N.D.	N.D.	N.D.	N.D.		
**Sessile *C. albicans* (*n* = 5)**	Fluconazole	EO *v/v%*	Fluconazole	EO *v/v%*	0.139 (0.038–0.250)	Synergy
	mg/L		mg/L			
	32 (32–>32) ^a^	6.25	0.5	0.09 (0.045–0.09)		
	Micafungin	EO *v/v%*	Micafungin	EO *v/v%*	1 (0.75–1)	Indifferent
	mg/L		mg/L			
	2	6.25	1 (0.5–1)	3.125		
**Sessile *C. auris* (*n* = 5)**	Fluconazole	EO *v/v%*	Fluconazole	EO *v/v%*	0.278 (0.275–0.5)	Synergy
	mg/L		mg/L			
	512 (512 –>512) ^b^	3.12 (1.56 –>6.25)	64 (64–128)	0.09 (0.04–3.125)		
	Micafungin	EO *v/v%*	Micafungin	EO *v/v%*	0.088 (0.043–0.75)	Synergy
	mg/L		mg/L			
	128 (32–128)	1.56 (1.56–3.125)	4 (4–8)	0.09 (0.02–1.56)		

N.D.: no data; ^a^ MIC is offscale at >32 mg/L, 64 mg/L (one dilution higher than the highest tested concentration) was used for analysis; ^b^ MIC is offscale at >512 mg/L, 1024 mg/L (one dilution higher than the highest tested concentration) was used for analysis.

## Data Availability

All data generated during the study are presented in this paper.

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
