# Peer review of "Evaluation of the Antimicrobial and Antivirulent Potential of Essential Oils Isolated from Juniperus oxycedrus L. ssp. macrocarpa Aerial Parts"

_microorganisms, 2022, doi:10.3390/microorganisms10040758_

Round 1

Reviewer 1 Report

The authors evaluated the antimicrobial and antivirulent potential of essential oils isolated from Juniperus oxycedrus L. ssp. macrocarpa against the variety of human pathogens.

In general, the manuscript is very well written, nicely introduced and provides very detailed methods section. The experiments are elegantly designed, with the aim to tackle different angels of pathogenicity and therefore understand better the mechanism of evaluated EO.

However, my only major concern is related to the discussion. The authors did test the potential antimicrobial therapeutic potential, but how exactly would this be applied? Systemic application of EO seems too unreal, so how would this really be used in practice? I would assume that it would rather have a local application or maybe used for the prevention of biofilm formation on cathethers or other materials relevant for the hospital environment. The authors mention in the last sentence the animal experiments - how exactly would you perform it?

Therefore, I think that is it crucial to put the realistic perspective of potential application in the discussion - otherwise, this paper would be one of the many similar papers that are testing tens of different EOs, but it does not more further than in vitro assays. 

Minor comments:

  • did the authors test isolated compounds from EO (e.g. alpha-pinene or limonene) to see if these are the ones responsible for the effect observed?
  • did the authors test the effect of EO on mixed species biofilm, often found in vivo?
  • few tecnical mistakes: species in italic (489-491), "pne" (line 506), "molceular" (line 626)

All in all, this is a very nice study that needs slightly stronger discussion in order to be acceptable for publication.

Author Response

The authors evaluated the antimicrobial and antivirulent potential of essential oils isolated from Juniperus oxycedrus L. ssp. macrocarpa against the variety of human pathogens.

In general, the manuscript is very well written, nicely introduced and provides very detailed methods section. The experiments are elegantly designed, with the aim to tackle different angels of pathogenicity and therefore understand better the mechanism of evaluated EO.

Dear Reviewer,

Thank you for taking the time to assess the suitability of our manuscript for publication in Microorganisms (MDPI). In addition, we are sincerely thankful for the positive attitude towards our paper, and the constructive comments provided by the Reviewer to further improve the quality of our manuscript.

However, my only major concern is related to the discussion. The authors did test the potential antimicrobial therapeutic potential, but how exactly would this be applied? Systemic application of EO seems too unreal, so how would this really be used in practice? I would assume that it would rather have a local application or maybe used for the prevention of biofilm formation on cathethers or other materials relevant for the hospital environment. The authors mention in the last sentence the animal experiments - how exactly would you perform it?

Therefore, I think that is it crucial to put the realistic perspective of potential application in the discussion - otherwise, this paper would be one of the many similar papers that are testing tens of different EOs, but it does not more further than in vitro assays.

Dear Reviewer thank you for this constructive comment and we have specified in the discussion the next practical aspect as a pharmaceutical formulation  ( lines 631-640).

Minor comments:

    did the authors test isolated compounds from EO (e.g. alpha-pinene or limonene) to see if these are the ones responsible for the effect observed?

Thank you for your question. At the present stage of this research, our primary aim was to assess the antimicrobial effect of the entire EO (the isolation of which is a novel finding by itself, from this Sardinian endemism), as the interaction between the individual components (listed in Table 1) may have profound effects on the biological potency of the EO. Nevertheless, the authors agree that mechanistic studies are warranted. In the future, more specific studies with the individual chemical constituents should be performed with the same model systems, to ascertain exactly, which constitutent modulates the antimicrobial activity of the isolated EO.

    did the authors test the effect of EO on mixed species biofilm, often found in vivo?

Thank you for your question. In the present (screening) stage of the research, only mono-species biofilms (S. aureus MSSA and MRSA for bacteria and Candida spp. for yeasts) were included as model systems in our methodologies. However, as the Reviewer pointed out, in realistic in vivo situations, microorganisms often reside in multi-species, mixed biofilms. Therefore, our future studies may include such model systems to attain experimental data more relevant to real-world situations.

    few tecnical mistakes: species in italic (489-491), "pne" (line 506), "molceular" (line 626)

Thank you for pointing out these errors. The mistakes listed have been corrected.

All in all, this is a very nice study that needs slightly stronger discussion in order to be acceptable for publication.

Once again, we are thankful for the Reviewer for providing constructive comments.

Reviewer 2 Report

u manuscript "Evaluation of the antimicrobial and antivirulent potential of essential oils isolated from Juniperus oxycedrus L. ssp. macrocarpa aerial parts" antibacterial, antifungal, anti-virulence and resistance-modulating properties of the EO of Juniperus oxycedrus L. ssp. macrocarpa aerial parts were evaluated. Comments follow:
When determining MIC and MBC, which antibiotic was used as a control? The same question applies to the antifungal activity of the compound.
What did the essential oil dissolve in? If it went directly into the broth, was there a separation of phases? If DMSO or another solvent was used, at what concentration was it used and was its antimicrobial effect tested?
Table 5. The title is wrong and needs to be corrected.

Author Response

u manuscript "Evaluation of the antimicrobial and antivirulent potential of essential oils isolated from Juniperus oxycedrus L. ssp. macrocarpa aerial parts" antibacterial, antifungal, anti-virulence and resistance-modulating properties of the EO of Juniperus oxycedrus L. ssp. macrocarpa aerial parts were evaluated. Comments follow:

Dear Reviewer,

Thank you for taking the time to assess the suitability of our manuscript for publication in Microorganisms (MDPI). In addition, we are sincerely thankful for the positive attitude towards our paper, and the constructive comments provided by the Reviewer to further improve the quality of our manuscript.

When determining MIC and MBC, which antibiotic was used as a control? The same question applies to the antifungal activity of the compound.

Thank you for your comment. During the MIC and MBC assays, some of the antibiotics listed for the MIC-reduction assay (namely ampicillin, ciprofloxacin, tetracycline, erythromycin and chloramphenicol for Gram-positive bacteria, while ampicillin, ciprofloxacin, tetracycline, gentamicin and chloramphenicol for Gram-negative bacteria) were used; negative control was phosphate-buffered saline (PBS). As the control strains utilized in this study are used extensively, their AST is consistently validated, and previous results were reported elsewhere both by our group and other researchers, we did not report every result from the MICs for the positive controls, only the relevant ones for the MIC-reduction assay. In the antifungal/fungal antibiofilm assays, fluconazole and micafungin were used as positive controls, respectively, with PBS being utilized as negative control.

What did the essential oil dissolve in? If it went directly into the broth, was there a separation of phases? If DMSO or another solvent was used, at what concentration was it used and was its antimicrobial effect tested?

Thank you for your question. The essential oil was dissolved in DMSO for analyses (this has been included in the MS), as the separation of phases was observed in cases, where the volume of the broth/medium was considerably higher than the volume of the essential oil. The solvent concentration was always <1 V/V% in bacterial and fungal assays, and always <2 V/V% in cell cytotoxicity assays. Owing to an extensive experience in biological screening assays, we have tested the biological effect of DMSO in these model systems as a solvent control, and found, that – when present in these small concentrations – the solvent has no effect on the results of the biological assays.

Table 5. The title is wrong and needs to be corrected.

Thank you for pointing out the error. The mistake has been corrected.

Once again, we are thankful for the Reviewer for providing constructive comments.

Round 2

Reviewer 2 Report

Although a sentence has been added to the manuscript on the use of DMSO as a solvent, I believe that the explanation given by the authors to me should be added to the text of the manuscript. The rest was made according to suggestions.

Author Response

Although a sentence has been added to the manuscript on the use of DMSO as a solvent, I believe that the explanation given by the authors to me should be added to the text of the manuscript. The rest was made according to suggestions.

Dear Reviewer,

Thank you for taking the time to assess the suitability of our manuscript for publication in Microorganisms (MDPI). In addition, we are sincerely thankful for the positive attitude towards our paper, and the constructive comments provided by the Reviewer to further improve the quality of our manuscript.

The information/sentences requested by the Reviewer were included in the manuscript at L156-L161.